# The Potential of Endophytes in Improving Salt–Alkali Tolerance and Salinity Resistance in Plants

**DOI:** 10.3390/ijms242316917

**Published:** 2023-11-29

**Authors:** Xueying Guo, Wanrong Peng, Xinyi Xu, Kangwei Xie, Xingyong Yang

**Affiliations:** 1College of Pharmacy, Chengdu University, Chengdu 610106, China; 13696470904@163.com (X.G.); pwr0530@163.com (W.P.); xuxinyi0669@163.com (X.X.); 13036699560@163.com (K.X.); 2College of Life Sciences, Chongqing Normal University, Chongqing 401331, China; 3Antibiotics Research and Re-Evaluation Key Laboratory of Sichuan Province, Chengdu University, Chengdu 610106, China

**Keywords:** salinity stress, endophytes, saline–alkali soils, salinity resistance

## Abstract

Ensuring food security for the global population is a ceaseless and critical issue. However, high-salinity and high-alkalinity levels can harm agricultural yields throughout large areas, even in largely agricultural countries, such as China. Various physical and chemical treatments have been employed in different locations to mitigate high salinity and alkalinity but their effects have been minimal. Numerous researchers have recently focused on developing effective and environmentally friendly biological treatments. Endophytes, which are naturally occurring and abundant in plants, retain many of the same characteristics of plants owing to their simultaneous evolution. Therefore, extraction of endophytes from salt-tolerant plants for managing plant growth in saline–alkali soils has become an important research topic. This extraction indicates that the soil environment can be fundamentally improved, and the signaling pathways of plants can be altered to increase their defense capacity, and can even be inherited to ensure lasting efficacy. This study discusses the direct and indirect means by which plant endophytes mitigate the effects of plant salinity stress that have been observed in recent years.

## 1. Introduction

Frequent climate variation and human activities have increased the intensity of various abiotic stressors, among which is saline–alkali stress, resulting mainly from soil-saline alkalization, is especially challenging [1,2]. Saline–alkaline soils usually featured high salt and pH (above 8.0) levels. Salt stress is mainly generated by Na_2_Cl, Na_2_SO_4_, and other neutral salts, whereas saline–alkali stress is generally caused by basic salts (e.g., NaHCO_3_, Na_2_CO_3_) [1,3]. The salinization of arable soil is gradually increasing globally and poses a considerable threat to sustainable agriculture [4]. According to the latest soil survey in China, saline soils cover approximately 36.933 million hm^2^; for residual saline soils, approximately 44.867 million hm^2^ are covered; for potential saline soils, 17.333 million hm^2^ are covered; and for all types of saline soils, 99.133 million hm^2^ are covered [5]. Saline soils are primarily characterized by copious amounts of soluble salts that hinder plant growth. Highly saline or alkaline conditions can severely damage plants to the point of death if not mitigated (Figure 1). The ongoing economic development and population growth complicates the maintenance of the arable-land limit of 120 million hm^2^. Additionally, the area covered by various types of saline land exceeds 140 million hm^2^; thus, the comprehensive utilization of saline land would considerably alleviate arable land and the food production crises. Saline land, an important land resource, must be urgently improved and utilized to promote food security and improved living standards [6].

Chemical, biological, physical, and agronomic improvements are currently the main technological approaches used to manage saline soils [7]. There are also soil-based improvement methods for these soils; however, biological improvements are mainly plant-focused [1,4]. One such example is when salt-tolerant crops such as sunflowers, licorice, alfalfa, and others are planted to improve soil structure [1,3,4]. Endophytes may contribute to the chlorophyll content of plants under salt stress, effectively increasing their photosynthetic capacity [8,9]. Concurrently, they can also absorb free salt ions from the soil, such as for sunflowers with sodium ions (Na^+^) [3]. In addition, these plants can become the main gene carriers used in saline-land management, in which endophytes can be extracted and widely used for genetically improving salt tolerance in various crops and ecologically engineered land-management plants [3,10]. 

In addition to the traditional treatment methods noted above, a microbial treatment using plant inter-rhizosphere bacteria and endophytes has recently attracted considerable attention among researchers, owing to its sustainability and ability to promote ecological conservation [11]. Endophytes have various functions such as plant growth promotion, xenobiotic degradation, pollution remediation, and bacterial chemotaxis, which can provide new ideas for saline-land management [12]. Microbial remediation approaches exhibit notable advantages in mitigating toxic environmental contaminants. These include a high reaction efficiency, low cost, and a lack of observed side effects [13]. A breakthrough in saline-soil treatments may therefore be possible through studying plant endophytes. Therefore, in this study, we focused on the interactions between plant endophytes and saline environments, and the mechanisms of endophytes in saline-land management. This paper discusses four aspects: the traditional means for saline-soil treatment, the probiotic effects of endophytes, the mechanisms of salt-stress mitigation using endophytes, and the application of endophytes for salt-stress mitigation. The aim of this study is to determine the best method to treat saline–alkaline soils in an environmentally friendly and sustainable manner.

## 2. Plant-Associated Endophytes

Plant-associated endophytes refer to all microorganisms, including bacteria, fungi, and actinomycetes, that inhabit the internal and intercellular tissues of plants throughout their entire or partial life cycle [14]. These microorganisms can colonize healthy roots, stems, bark, leaves, petioles, flowers, fruits, and seeds without causing any apparent harm to, or pathogenic infection within, the host plants [15,16]. Endophytes form internal connections with host plants through seed transmission and benefit from obtaining nutrients [17].

Endophytes are used in agricultural production and perform nitrogen fixation, promote plant growth, and help plants withstand adversity [12,18,19]. They can effectively contribute to agricultural production, such as through the production of biofertilizer to promote plant growth [18], thus contributing towards the degradation of organic pollutant residues [20] by preserving and maintaining freshness, and controlling pests and diseases [21,22]. Endophytes can effectively reduce the content of organic pollutants in plants and facilitate the efficient use of soil to produce safe agricultural products [23]. Endophytes can enhance a plant cell’s ability to dissolve minerals and metals by secreting low-molecular-weight organic acids and siderophore-like metal-specific ligands, thereby altering soil pH and enhancing binding activities [19,24,25]. They can also be applied to the management of heavy-metal land pollution [26], organic and atmospheric pollutants, volatile organic compounds, inorganic substances, water pollution, and other ecological pollutants addressed through saline-land management [20]. Endophytes can also synthesize diverse chemical components through various biological activities, and these components are expected to become candidates for new drugs [27,28,29].

## 3. Plant Salt Stress and Mechanism Underlying Salt Tolerance in Plants

High concentrations of salts, especially Na^+^, within soils can cause salinity damage or salinity stress in plants; this interferes with their normal growth and development, resulting in agricultural losses [30]. Damage to plants in saline soils includes the loss of chlorophyll, decreased photosynthesis rates, reduced cell division, reactive oxygen species (ROS) production, inactivation of antioxidants, and altered plant hormone biosynthesis and signaling, which may in turn lead to reduced plant survival rates and yields [31,32,33]. The state of Chinese arable land is currently dire; food and arable land scarcity is an increasingly urgent concern. The management of saline land in mitigating salt stress and producing sufficient food and resources to ensure food security has become a top priority [32].

Plants have evolved a complex set of response mechanisms to adapt to harsh high-salt environments [34]. Plants adapt to salinity mainly by regulating the transcription of gene networks involved in ion transport, osmotic balance, ROS scavenging, and phytohormone regulation, as well as post-translational modifications and epigenetic factors [35,36]. Under saline conditions, plant roots first sense the osmotic and ionic stresses of salinity, and then rapidly respond to altered signaling. These signaling events depend on secondary messengers such as calcium ions (Ca^2+^), phosphatidyl inositol, ROS, and phytohormones [3,33,37]. A typical example is the model of the NaCl response mediated by calcium signaling that was proposed by Deinlein [38]. A transient increase in intracellular calcium-ion concentration, which triggers downstream signaling pathways, occurs under NaCl stress and induces a momentary increase in intracellular Ca^2+^ signaling, altering the transcriptional profile of Na^+^ transporter protein genes, including *HKT1*, *NHX*, and *SOS1*. ROS signaling induces antioxidant mechanisms that protect organisms from oxidative stress (Figure 2) [3,4,39,40].

However, plant resistance is strongly limited, and high salinity continues to be a major abiotic stress limiting crop growth and production. Certain cash crops are especially vulnerable to yield reductions in difficult soil environments; this phenomenon has resulted in a large amount of land in China having no practical use, thereby creating resource wastage. Even land that was once highly productive is no longer suitable for growing crops owing to its high salinity. This necessitates manual intervention to ensure optimal use of vacant land.

## 4. Endophytes and Management of Salt Stress

### 4.1. Probiotic Effects of Endophyte

There are numerous rich and diverse sources of endophytes. Soil is the main source of endophytes [41,42], but the air in which plants grow [43], phytophagous insects [44], and plant seeds are also sources of endophytes [45]. This is precisely because endophytes originate from the environment where they grow and live, and have become a natural part of the plant microecosystem through long-term co-evolution with plants. They can promote the adaptation of plants to harsh environments and strengthen the ecological balance throughout the ecosystem [46]. Endophytes from saline environments naturally adapt to saline environments and can help plants to more effectively adapt to saline environments, thus promoting the management of saline lands [47,48].

Endophytes form symbiotic or pathogenic relationships with plants. Their beneficial effects on plants include promoting plant uptake of nutrients, such as nitrogen, phosphorus, and ions; regulating plant growth and development by regulating plant hormones (growth hormone, cytokinin, ethylene, etc.); helping plants resist biotic or abiotic stresses; synthesis of substances such as catalase and superoxide dismutase to prevent the harmful effects of ROS; synthesis of small-molecule osmolytes, such as alglucan and extracellular polysaccharides, to improve plant water content and protect plant cells from water loss and stabilize soil aggregates; and synthesis of antibiotics to protect plants from the threat of some pathogenic bacteria [12,18,49,50,51,52]. Endophytes simultaneously take direct and indirect actions. These probiotic effects not only remove external enemies, such as harmful phytopathogenic bacteria or fungi, but they also provide beneficial mineral nutrients for growth, while creating a relatively stable growth environment for plants [51,52]. Different plants grow under adverse conditions and exhibit certain survival abilities (Figure 3) [53]. 

### 4.2. Endophytes for Saline-Land Management

The management of endophytes in saline soils can be elucidated through studying their origins and beneficial effects. Endophytes originate from the environment, and the number and structural distributions of endophyte species and communities are influenced by the plant’s environment. Plants growing in saline soils have more endophytes with a better salinity resistance. Interestingly, a unique saline plant ecosystem exists in saline soils, where the salophytes in the system grow in saline soils, and improve the saline-soil environment [54]. These halophytes can grow and complete their life cycle in habitats with ion concentrations above 200 mM [55]. As an example, the endophyte diversity of the saline organism *Heterocarpus* was positively correlated with the salt stress gradient, but different tissues were more sensitive to the salt gradient than different developmental stages. The results showed that *Heterocarpus* tissues were more dependent on the symbiosis of endophytes when subjected to salt stress. Notably, the more saline in the soil, the richer the endophyte diversity, and the greater their effect on the plant. Therefore, endophytes produced in such environments are warranted thorough research, and potentially present numerous prospects for biological soil treatments [56,57,58]. Endophytes inhabiting plants in saline environments can be isolated and used in the bioremediation of land damaged by high-salinity conditions.

Therefore, saline organisms such as endophytes have been studied extensively. A substantial number of studies have focused on the endophytes in crops growing in saline environments; this is notable because crop harvest is related to the global food problem, and feeding people in all countries is a constant issue. Moreover, the extraction of endophytes from saline plants to improve saline lands is feasible and represents an important research topic given that plants that grow in saline lands are resistant to salinity.

## 5. Mechanisms Whereby Plant Endophytes Mitigate Salt Stress

### 5.1. Hazards of High Salt Penetration to Plants

Soil salinity severely affects plant growth and development and causes unavoidable losses in crop productivity. The damage caused to plants by saline soils is a direct result of salinity stress. Generally, the numerous mechanisms underlying plant damage by salt or alkalis are the same. Therefore, stress-reducing solutions may substantially improve highly saline and alkaline soils. Plant damage from salinity stress can be categorized into osmotic stress, ion toxicity, and a high-pH environment caused by alkaline stress, which affects seed germination, growth and development, and plant gene expressions to varying degrees [59].

The osmotic and ionic imbalances in plants caused by salt stress result mainly from alterations of the Na^+^ and potassium ion (K^+^) contents [60]. The aggregation of these ions substantially alters their potential balance in the plant, which affects the structure of the plasma membrane of cells. This in turn demonstrably increases permeability, leading to nutrient loss and triggering different degrees of salt-ion toxicity [61]. In addition, heavy-metal ions can enter plants through the root system when soil salinity increases significantly, thereby triggering heavy-metal toxicity. This can affect plant growth and development, not only by reducing yields, but in severe cases causing death [62,63].

In contrast, alkaline stress subjects plants to high-pH environments that severely inhibit plant growth, as a high-pH significantly hinders root development. The root system is most directly and primarily affected because of its direct contact with this adversity stress [3,64]. pH also causes an ion imbalance near the root system through accumulating metal ions and phosphorus precipitation around the root system [3,59]. This affects the uptake of nutrient elements, resulting in reduced plant root vigor and decreased root absorption function, and leading to the wilting of aboveground leaves, affecting normal photosynthesis; this disrupts plant growth metabolism or hinders physiological functions [65]. This process also inhibits the germination of plant seeds, resulting in long germination times, reduced germination rates, and heterogeneous shoots (Figure 4).

Osmotic stress interferes with plant nutrient uptake. Both salinity and alkalinity stress reduce the water potential, thus complicating water uptake by plants, and increasing the osmotic potential of the soil solution, thereby causing osmotic stress in plant cells. Osmotic stress in rice subjected to high-soil-salinity and high-sodium content levels substantially changes the concentration of Na^+^ and K^+^ in rice cells, thereby reducing the permeability of the cell membrane, hindering nutrient uptake, and affecting plant metabolism and development [66,67].

### 5.2. Direct Action of Endophytes

Soil affects plant growth and can introduce osmotic stress to plants. Therefore, directly reducing ions in the soil that cause hyperosmotic conditions is a direct solution to this problem. *Arbuscular mycorrhiza* (AM) can mitigate the negative effects of salt and heavy-metal stress on the growth of small-fruited white spurges by promoting nutrient uptake, regulating the ion balance in plants, and increasing Na^+^ and Cd uptake rates [16,68]. Therefore, some salt-tolerant plants can remove large amounts of salt from saline soils, and endophytes play an important role in this process. Therefore, their importance in ecology is gaining increasing attention [16]. Certain endophytes secrete organic acids, such as formic, acetic, propionic, and glycolic acids, to lower soil pH and facilitate saline-land management [69]. Under high salinity or pH, salt-loving microbes produce specific metabolites that are active in biodegrading and remediating the environment. Therefore, salinophilic or salt-tolerant microorganisms with ecological remediation abilities isolated from saline environments can be effectively used to mitigate saline pollution [70].

The famous salt-tolerant-pioneer plant, alkali poncho can improve coastal salinity [71]. There is a relatively rich diversity of moderately salinophilic bacteria and phylogenetic diversity in *Salicornia salsa*, and numerous new microbial taxa are latent [72,73]. Xi et al. isolated and identified salinophilic bacterial strains from the roots and leaves of salt pondweed (*Suaeda salsa*) and rice grass (*Spartina anglica*) and observed an auxotrophic effect on the weathering of salt minerals [74]. Their 16SrRNA sequence analysis showed that this strain is probably *Planococcus*, which can secrete 1-aminocyclopropane-1-deaminase with phenanthrene and pyrene as the only carbon sources. Moreover, it can perform phosphorus solubilization and IAA production, which can degrade phenanthrene and pyrene in soil, as well as potentially promoting growth. In addition to the isolated microbes, bioremediation experiments on coastal saline soils indicated that the species and number of microbes in the inter-rooted soil of the planted area exceeded those in the bare soil of the same geographical area [50,75]. This may have resulted from these plants improving saline soils or the interaction between the plants and the inter-rooted microbes to promote the growth and reproduction of degradative microbes [76]. Endophytes promote salt decomposition, thus reducing the risk of saline soils and consequently promoting the restoration of saline soils [77].

### 5.3. Indirect Action of Endophytes

In certain cases, even in instances where an endophyte cannot directly absorb or promote the decomposition of harmful substances in saline land, saline land can nonetheless be managed if the plants are more salinity resistant and can grow unproblematically in saline land. Endophytes can cause plants to grow in saline land, mainly because they enhance plant resistance to salinity.

Regulation of phytohormone biosynthesis and signaling pathways, including indole acetic acid, gibberellic acid, oleuropein lactone, abscisic acid, and jasmonic acid, is the mechanism by which endophytes induce stress tolerance [78,79]. These pathways accumulate osmoprotectants such as proline, glycine betaine, and sugar alcohols, and regulate ion transport proteins such as SOS1, NHX, and HKT1 [78,80]. At the genetic level, salt-tolerant endophytes induce the expression of salt-responsive genes through various transcription factors and post-transcriptional and post-translational modifications [31].

These mechanisms are demonstrated through the ability of endophytes to promote plant growth by secreting various substances that accelerate the availability of mineral nutrients; aiding the production of plant hormones, iron carriers, and enzymes; and activating systemic resistance to plant pests and pathogens. Certain endophytes can dissolve insoluble phosphate and potassium salts into phosphorus and potassium salts that can be absorbed and used by plants, thereby increasing the content of fast-acting soil phosphorus and potassium and promoting plant growth and development [81]. Specifically, endophytes can improve physiological and biochemical levels; promote the secretion of various substances to help plants minimize salt stress; enhance the uptake of nutrients from the soil to promote plant growth and development; and promote the synthesis of osmolytes, stress responses, and ion transporter genes in the host. This increases the stress response of the host to saline environments, thus improving host resistance and crop yields in saline environments [82].

#### 5.3.1. Accumulation of Plant Hormones

Plant endophytes also contribute to phytohormone synthesis. Phytohormones are organic substances that are metabolically synthesized by plant cells in specific plant tissues in response to specific signals, and bind to specific protein receptors to regulate plant growth and development [6]. Among these phytohormones, gibberellin, cytokinin, and oleuropein sterol promote growth, whereas abscisic acid and ethylene inhibit growth. Endophytes increase resistance to salinity by increasing the synthesis of growth promoters and decreasing that of growth inhibitors.

Intergenic bacteria promote plant growth, as well as increasing their induced system to achieve resistance to salinity stress through various processes, such as antioxidant enzyme activity and reduction of ethylene levels through ACC deaminase activity [83]. In addition, ten strains of salt-tolerant bacteria were screened and evaluated for their PGP characteristics using plant–microbe interaction tests under indoor and natural conditions. GC-MS analysis of the metabolites of the selected strains confirmed the presence of indole-like growth-inhibitory compounds, such as indole, indole-3-butyramide, benzylmalonic acid, and 4-methyl-2-pyrrolidone. These compounds also produced a demonstrable salt-tolerance effect following an inoculation of cotton [84]. Similar results were obtained by other researchers who used the salt-tolerant endophytic bacterium *Enterobacter ludwigii* B30 to increase fresh and dry weights; carotenoid and chlorophyll contents; catalase and superoxide dismutase activities; indoleacetic acid contents; and K^+^ concentration. Without the *E. Ludwigii* B30 treatment, the malondialdehyde, proline, PSII [Y (NO) and Y (NPQ)], 1-aminocyclopropane-1-carboxylic acid, and abscisic acid contents of dogbanes decreased under salt stress [85].

#### 5.3.2. Accumulation of Osmoprotectants

Throughout the long-term evolution of plants, endophytic organisms living in hyperosmotic environments have developed unique and complex osmoregulatory systems, particularly bacteria, to adapt to the external hyperosmotic environment. Bacteria usually accumulate small-molecule substances in their bodies as osmoprotectants [86]. Their main components are low-molecular-weight substances such as sugars, alcohols, amino acids, and amino acid derivatives. The accumulation of this class of substances is a protective response of bacteria to the hyperosmotic environment, which can increase the osmotic potential of a cell to counteract external osmotic pressure, thus reducing water loss, protecting the cell from dehydration, and acting as a stabilizer and protector of the structures and functions of macromolecules in the cell.

Inoculation of rice with S. endophyticus OsiLf-2 resulted in the production of abundant osmotic-stress substances, including proline, polysaccharides, and exocytin, thereby enhancing the osmoregulatory capacity of rice and increasing its resistance to salinity [82]. Coincidentally, peanut co-evolved endophytes, such as Bacillus J22N and Bacillus REN51N, can regulate relative water content and increase osmoprotectant accumulation, thereby mitigating high-salinity stress and increasing yield in saline lands [87]. Guo et al. investigated the growth and physiological responses of the wetland plant, *Suaeda salsa*, inoculated with two endophytic bacteria, *Sphingomonas prati* and *S. zeicaulis*, from saline lands [88]. PCA, cluster analysis, and PLS modeling revealed two mechanisms by which *S. prati* enhanced the growth of *S. zeicaulis* to regulate plant salt tolerance. Conversely, *S. prati* increased intracellular osmotic metabolism, and *S. prati* promoted the production of CAT, an antioxidant enzyme system, and maintained permeability [88]. In four endophyte strains (N, L, K, and Y) from Phalaris, a 16S rRNA gene-sequence analysis showed that these strains belonged to genera *Pseudomonas, Bacillus*, *Mucor*, and *Mucor*, respectively. *Mucilaginibacter* and *Rhizobium* partially lowered salt stress by regulating osmoregulatory substances and antioxidant enzymes [19]. Chen et al. found that the endophyte *Epichloë bramicola* improved the salinity tolerance of wild barley under salt stress, likely because *E. bramicola* affects polyamine metabolism. Endophytic fungal osmotolerance genes expressed in plant cells and bred crops tolerant to hyperosmotic environments have attracted considerable attention from researchers [89].

#### 5.3.3. Regulation of Ion Transportation

Plant endophytes regulate ion-transport proteins. Mineral elements are important components of plants, as they help regulate important physiological and biochemical reactions, and maintain normal physiological activities in plants [90]. In the soil, only soluble inorganic ions can be directly absorbed and utilized by plant roots, whereas most mineral elements are mainly in an insoluble form and cannot be absorbed and utilized by plants. Plants absorb ions mainly through transporter proteins in the root epidermal cell membranes. Different transporter proteins have different affinities for ions and regulate ion uptake at different concentrations. Endophytes can influence the secretion of transporter proteins and thus affect the uptake of ions in a saline environment, ensuring that ions inside and outside of plant cells can reach an equilibrium state and improve the salinity tolerance of plants [91].

In addition, several microorganisms can improve the salt tolerance of plants by regulating ion-transporter genes. Guo et al. demonstrated the protective role of the salt-tolerant PGPR strain *Dietzia natronolimnaea* STR1, which regulated the expression of genes related to the SOS pathway (*SOS1* and *SOS4*), vacuolar transport (*NHX1*), potassium ion transport (*HAK* and *HKT1*), and antioxidant enzymes (APX, MnSOD, CAT, POD, GPX, and GR) in NaCl-stressed wheat plants [92]. The application of another bacterial strain, *Burkholderia phytofirmans*, to the salt-stressed *Arabidopsis* plants induced the expression of ion-transporter genes, specifically *HKT1*, *AKT1*, *NHX1*, and *SOS1* [93]. The upregulation of other salt-responsive genes (*bZIP* (*BZ8*) and *GIGANTEA* (*GIG*)) as well as transporter genes (*SOS1* and *NHX1*), was observed in rice plants inoculated with the salt-tolerant PGPR *B. aryabhattai* MS3; this upregulation was associated with enhanced salt tolerance in the rice plants [94,95]. Endophytic fungi also induce the transcription of gene-encoding ion transporters [96]. In this context, the inoculation of salt-stressed *Arabidopsis* plants with *P. indica* also confirmed the ability of the endophyte to induce the expression of *HKT1* (high-affinity potassium transporter 1), *KAT1*, and *KAT2* (inwardly rectifying K+ channels), which help regulate Na+ levels in plant tissues [97]. *Rhizobacteria* and *Clumping mycorrhizal* fungi increased the resistance of tall fescue to salinity stress, as well as increased aboveground and root biomass, nutrient uptake (organic carbon, total nitrogen, and total phosphorus concentrations), and accumulated K^+^, while decreasing the Na^+^ concentration [91]. Lanza et al. also found that the endophytic bacterium *Serendipita indica* reduced the sodium content in *Arabidopsis* plants exposed to salt stress [61].

#### 5.3.4. Salt-Responsive Gene Expression

Numerous studies have identified the roles of microorganisms in enhancing the salinity tolerance of plants, including promoting the expression of plant genes associated with photosynthesis, ROS scavenging, osmolyte accumulation, ion homeostasis, and phytohormone signaling. However, in many cases, the specific underlying mechanisms remain unclear. Therefore, there is an urgent need to study endophytes that regulate salt-responsive genes in plants [98,99]. Plant endophytes can alter original gene expressions in plants. At the genetic level, endophytes can alter the salt tolerance of plants from the basal level by upregulating or modifying plant genes for material and nutrient uptake.

Several bacterial strains, including *Arthrobacter volvulus*, *Microbacterium oxidans*, *A. aureus*, *Bacillus* sp., and *Pseudomonas* sp., tolerated salt stress by promoting the expression of plant genes related to photosynthesis, ROS scavenging, osmolyte accumulation, ion homeostasis, and phytohormone signaling [100]. An inoculation of *A. thaliana* with *Burkholderia pseudomallei* PsJN stimulated the upregulation of the ABA signaling genes *RD29A* and *RD29B* (relative to drought), whereas the expression of the jasmonic acid (JA) biosynthesis gene lox2 (lipoxygenase 2) was downregulated [93]. This suggests that the ability of the strains to enhance salt tolerance occurs mainly by affecting plant ABA signaling and response pathways. Dong et al. isolated two strains of the endophyte SYSU 333322 and SYSU 333140 from saline plants [101]. The 16S rRNA gene-sequence analysis showed that these two strains belonged to the endophytic genus *Arthrobacter* and harbored potassium-ion uptake-related genes; thus, they enhanced the ability of *Arabidopsis* to uptake and secrete different compounds, ultimately enhancing salt tolerance at the genetic level. Similarly, applications of salt-tolerant strains such as *B. aryabhattai* H19-1 and *B. mesonae* H20-5 have demonstrably stimulated ABA biosynthesis genes in tomatoes, thereby improving plant performance under salt stress [102]. The selective inoculation of sensitive varieties with endophytic strains of salt-tolerant varieties can also increase salt tolerance. For example, salt-tolerant *Fusarium oxysporum* isolated from a salt-tolerant variety of *Pokkali* rice successfully transformed salt-tolerant traits into a sensitive variety of *ir64* [103]. The present study shows that endophytic inoculation induced the upregulation of 1348 genes, including receptor proteins, signal-transduction proteins, secondary metabolites, and transcription factors, which are essential for improving stress responses in sensitive cultivars (Table 1).

## 6. Application of Endophyte in Mitigating Salt Stress

### 6.1. Application of Single Endophyte

The wheat seedlings treated with an endophyte, *Pantoea agglomerans* YN1, from healthy wheat stems showed significant increases in plant height and root length; chlorophyll, carotenoid, and proline contents; CAT, POD, and SOD activities; and a significant decrease in malondialdehyde content under 150 mM NaCl stress, which demonstrated the potential of YN1 in promoting plant salt tolerance [104]. Manjunatha et al. found that the endophytic fungi improved salt tolerance in wheat at the seedling stage [105].

In recent years, transferring high-grade endophytes to organisms to increase the range of endophytes in promoting plant salinity tolerance and increasing the biological resource pool has become an increasingly common practice. By comparing the proline and MDA contents of wheat seedlings inoculated with endophytes extracted from soybeans 252 and 254, Xu et al. found that both strains successfully inhibited and repaired salinity damage to a certain extent [106]. Moreover, both reduced the degree of cell membrane lipidation in seedlings, which improved the survival rate and salt resistance of wheat. The *Zea mays* endophyte can repair the growth of wheat seedlings under NaCl (150 mmol/L) stress, but the growth and reproduction of this endophyte can be inhibited under high-salt (NaCl 300 mmol/L) conditions [107]. Lei et al. inoculated the endophytic bacterium PP04 of the genus *Panicum* from the roots of hybrid wolfsbane under high, medium, and low salt stress, and found that under different concentrations of salt stress [108] the endophytic bacterium PP04 promoted the growth and development of hybrid wolfsbane by inducing different antioxidant protective enzyme activities in the hybrid wolfsbane. This decreased membrane lipid peroxidation and decreased the malondialdehyde (MDA) content. Lei et al. also selected three inter- and endophytic strains with a strong salinity-stress tolerance, inoculated them into the roots of alfalfa seedlings, and found that they could substantially improve the salinity tolerance and growth efficiency of alfalfa [109]. Wang obtained seven bacterial strains from the rhizospheres of alkali ponies that were beneficial to *A. thaliana* and wheat under salt stress [110].

### 6.2. Combined Application of Multiple Endophytes

The study findings above show that combinations of individual bacteria can be used to help plants resist external stressors and thrive. In most cases, however, multiple beneficial endophytes inoculated simultaneously on plants are more efficient than a single inoculation, thereby facilitating land salinization management or enhancing salt tolerance in plants [79]. Jha et al. also studied the effect of endophytes (*Pseudomonas pseudoalcaligenes*) on plant salinity tolerance by inoculating the rice variety GJ-17 with endophytes. This bacterium significantly increased crop yields under salt stress by synthesizing large amounts of betaine-like quaternary ammonium salts. The authors also noted that their combination (*P. pseudoalcaligenes + B. pumilus*) provided better protection for crop growth by inducing the synthesis of osmoprotectants and antioxidants better than a single application of *P. bifidum* or endophytic pseudoalkaloid-producing *Pseudomonas* [111]. Effects of mycorrhizal fungi (AMF), *Penicillium fuiculosum* (PF), and *Fusarium oxysporum* (FO) on wheat growth under saline conditions were studied. Wheat inoculated with AMF or PF+FO alone showed significantly lower yield enhancements in a saline environment than with an AMF and PF + FO co-inoculation, which enhanced yields by up to 43% [112].

## 7. Conclusions

Exogenously applied endophytes are considered an effective modern approach to improve plant growth within a saline environment. This method resolves the salinity-stress problem without causing any ecological pollution, thus demonstrating its environmental sustainability. The endophytes’ regulatory role is more often a means to reinforce the cultivation or growth process of plants through strengthening their ability to resist salinity during the growth process, and, in some cases, to absorb salinity.

However, it is currently only possible to apply these endophytes to plants that absorb salt from the soil. This process is complex, requires a certain level of expertise, and is labor and resource intensive. Furthermore, specialized ecological cultivation is time-consuming, which to a certain degree delays agricultural production and can ultimately produce yields and improved lands whose economic value are far below the cost of the input. Although endophytes have certain auxiliary effects in managing salinization, diversity has been observed in the types of saline-land pollution in China, including coastal saline land, inland arid saline land, saline land polluted by metals, and saline land polluted by hydrocarbon substances. Saline lands with different causes and compound pollutants require additional specialized research, and endophytes to produce different applications. In addition, microbial regulation is not a single regulatory entity; the microbial systems inside a plant, as a whole harmoniously, have a direct and close influence on each other, and certain studies have found that composite applications of multiple bacteria to plants has a better effect on plant growth. Therefore, there are still several specific mechanisms of action that are not yet clear, and certain researchers speculate that this results from the methylation of plant DNA. These mechanisms are worth exploring and require joint efforts from various disciplines.

Although the treatment of saline soils with endophytes is a promising environmentally friendly agricultural application, certain studies have shown that the results of field experiments are poor; therefore, considerable work remains to be conducted in progressing from laboratory tests to actual field production applications. Concurrently, relatively stable sources of endophytes are needed from salinity-resistant plants or crops isolated from endophytes to more effectively apply endophytes in managing soil salinization. These sources must be properly frozen, as they are extremely valuable assets. Future related research should ultimately examine (1) how endophytes can assist non-saline plants in absorbing and decomposing salts in the soil, (2) the treatment mechanisms and applications of endophytes in saline soils subjected to pollution from various substances, (3) the collaboration mechanisms of endophytes and other microorganisms and their applications, and (4) mechanisms for optimizing the effects of endophyte treatment in field applications.

## Figures and Tables

**Figure 1 ijms-24-16917-f001:**
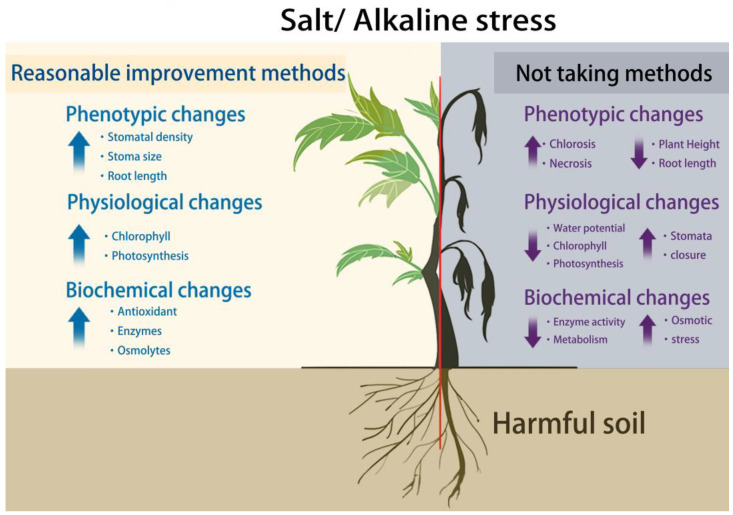
Schematic of the stresses for plants under high-salinity or high-alkaline growth conditions and their corresponding responses. With appropriate measures, such as irrigation, fertilization, and colonization with microorganisms, the plant’s height and root length can be increased; the photosynthetic rate can be strengthened by increasing chlorophyll content; and an increase in the number of some antioxidants and enzymes can help the plant resist adversity (**left**). If no reasonable measures are taken, the plant will become shorter in root length and height and will be at much higher risk of chlorosis; it will have less chlorophyll and its stomata will close, so its photosynthetic rate will decrease; it will be subjected to higher osmotic stress; and its metabolism will slow down. If nothing is done, the plant will probably die (**right**).

**Figure 2 ijms-24-16917-f002:**
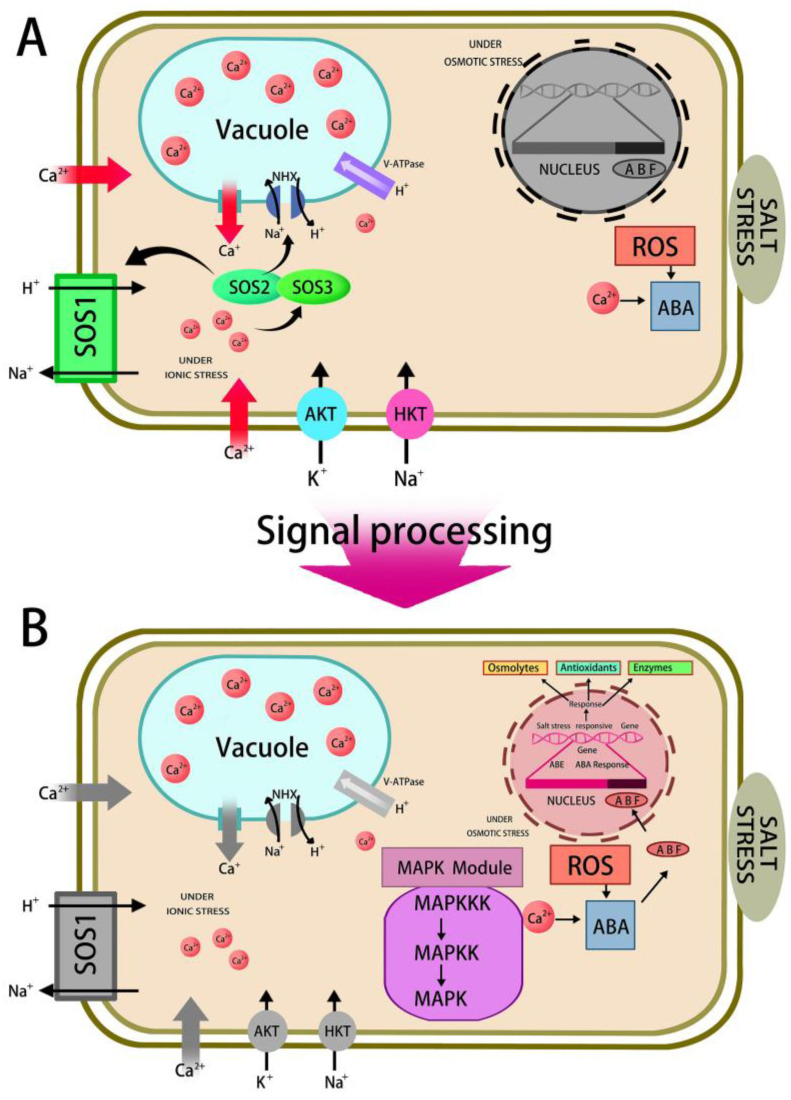
Schematic of the NaCl stress response mediated by calcium signaling in a plant cell. Hormonal signaling regulates physiological responses and activates several genes that respond to stress. (**A**) The ion homeostasis and osmotic stress-signaling pathways in plants are activated by salt-stress sensed by receptors. The SOS pathway maintains the cellular Na^+^ concentration either by sequestering it in the vacuole or by effluxing it from the cell. Cellular Na^+^ increase is associated with a sharp rise in Ca^2+^ level followed by its binding to SOS3. SOS3 activation of SOS2, positively regulates SOS1 and vacuolar NHX activity. In the non-stressed state, ABI inactivates SOS2, which can be degraded by proteasomal activity under salt stress. (**B**) ROS generation activates MAPK cascades and ABA-responsive genes, which are responsible for generating several osmoprotective and detoxifying proteins involved in salt tolerance. ABA: abscisic acid; ABF: ABA-responsive element-binding factors; AKT: protein kinase B (serine/threonine kinase); MAPK: mitogen-activated protein kinase; MAPKK: mitogen-activated protein kinase kinase; MAPKKK: mitogen-activated protein kinase kinase kinase; NHX: Na^+^/H^+^ exchanger; HKT: potassium transporter with high affinity; and SOS: salt oversensitivity.

**Figure 3 ijms-24-16917-f003:**
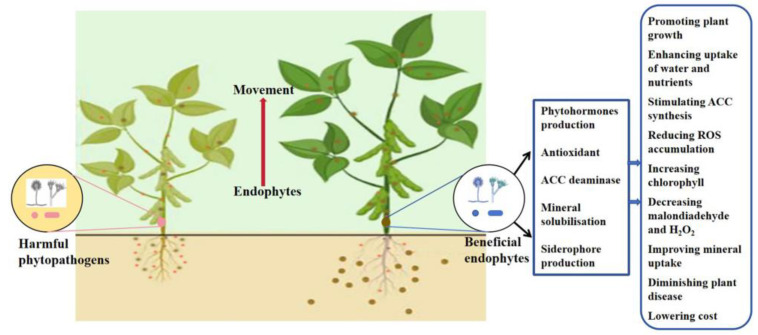
Model of beneficial endophyte spread into the soil. Endophytes promote the uptake of nitrogen, phosphorus, and potassium by the plant’s inter-roots, leading to increased root vigor and promoting growth in roots, stems, and leaves. They also have an antibacterial effect. This protects the plant from invasion by other plant pathogens, forming a natural “invisible protective shield”. Where ROS is reaction oxygen species; H_2_O_2_ is hydrogen peroxide; and ACC is 1-aminocyclopropane-1-carboxylic acid.

**Figure 4 ijms-24-16917-f004:**
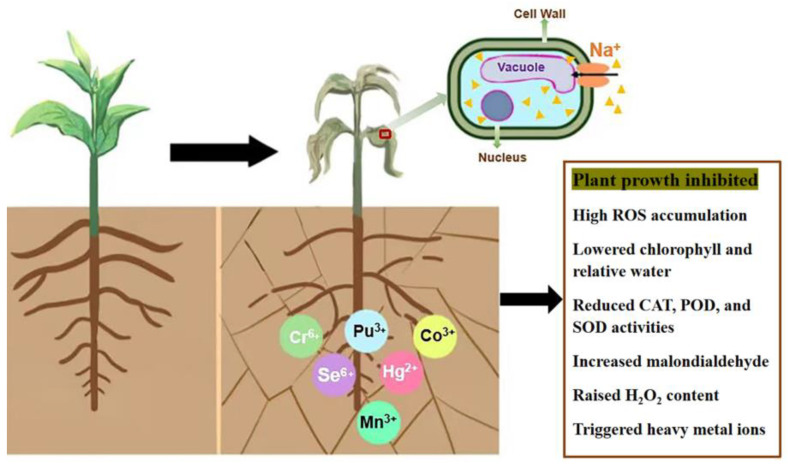
Diagram of plant transitioning from a healthy to a wilted state after being subjected to salt–alkali stress. When plants are exposed to saline stress, both intracellular osmosis and ionic imbalances are disrupted. Large amounts of sodium ions enter the plant cells, causing the vesicles to lose water and collapse and the plant to wilt. This leads to reduced root absorption and, in the worst case, plant death. Where ROS is reactive oxygen species; SOD is superoxide dismutase; CAT is catalase; POX is peroxidase; and H_2_O_2_ is hydrogen peroxide.

**Table 1 ijms-24-16917-t001:** The mechanisms of action of different endophytes in combating salinity stress.

Edophyte	Host	Mechanism	References
*Funneliformis mosseae*	*Nitraria sibirica*	It alleviates salt and heavy-metal stress by promoting nutrient absorption, regulating ion balance, and affecting Na+ and Cd absorption in plants.	[16]
*Bacillus japanicum*	Soybean, wheat	Reduce ethylene levels to achieve resistance to salt stress.	[83]
*B. cereus, B. subtilis,* *B. paramycoides,*	Cotton	Synthesize indole, indole-3-butylamide, benzyl malonic acid and 4-methyl-2-pyrrolidone.	[84]
*Enterobacter ludwigii*	*Cynodon dactylon*	The content of indoleacetic acid was increased, and the content of abscisic acid was decreased under salt stress to the host.	[85]
*Streptomyces albidoflavus*	Rice	Help rice produce rich osmotic-pressure substances, including proline, polysaccharide, and exotin, to increase the osmoregulation ability of rice.	[82]
*Bacillus firmus;* *B. tequilensis*	Peanut	Enhance accumulation of proline, reduced level of phenol and H_2_O_2_, and enhanced uptake of potassium.	[87]
*Sphingomonas prati,* *S. zeicaulis*	*Suaeda salsa*	Improved intracellular osmotic metabolism and promoted the production of CAT in the antioxidant enzyme system and retained permeability.	[88]
*Bacillus mobilis,* *Rhizobium jaguaris*	*Arabidopsis thaliana*	Regulate osmolytes and antioxidant enzymes.	[19]
*Claroideoglomus etunicatum*	*Lolium arundinaceum*	Increase shoot and root biomass and nutrient uptake (organic carbon, total nitrogen, and total phosphorus concentration), and accumulate K^+^, while decreasing Na^+^ concentration.	[91]
*Serendipita indica*	*Arabidopsis*	Produce a reduction in Na^+^ content in the plant roots and upregulation of chlorophyll a reductase.	[61]

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
