# Peer review of "The Potential of Endophytes in Improving Salt–Alkali Tolerance and Salinity Resistance in Plants"

_ijms, 2023, doi:10.3390/ijms242316917_

Round 1

Reviewer 1 Report

Comments and Suggestions for Authors

Guo et al. have reviewed the potential use of plant endophytes to deal with the salinity stress in plants grown in salt-alkali soils. However, the manuscript requires substantial improvements for clarity and readability. Below are my suggestions to address the issues raised:

1.      What do the authors mean by salt-alkali soil? No information is provided, and this term is not consistently used in the manuscript. This term needs clarification and consistent usage throughout the manuscript.

2.      The manuscript contains sentences that lack clarity and may appear to be AI-generated. Extensive language editing is required to enhance readability. For example: For example,

Line 26: "However, large tracts of land are wasted…" -> Rewrite for clarity.

Line 61-62: "In addition to traditional treatments such as physical, chemical, and agronomic improvements by plants…" -> Clarify the meaning.

Line 309: "Healthy to wilted plants after receiving salt-alkali stress?" -> Rewrite for clarity.

Line 546: "Endophytes are an emerging method for salinity treatment?" -> Clarify the statement.

3.      Some of the text needs reference. Authors should carefully check the whole text for this issue.

I am providing a couple of examples-

Line 43- Sunflowers, licorice, alfalfa, and other salt-tolerant plants can be planted (provide reference). Endophytes may contribute to the chlorophyll content of the plant under salt stress, effectively increasing their photosynthetic capacity (provide reference).

4.      All the figures are too general and do not provide any insights into the topic. Authors should refer to the relevant literature and update the figures with more meaningful content. For instance, Figure 1 and Figure 2 should include an overview of salinity and alkali stress sensing, signaling, and response in plants. For instance, Figure 2 is primitive, and several aspects of salinity stress are missing.

5.      Also, Figure legends are inadequate and ensure that figure legends provide sufficient information for clarity.

6.      Some sections contain unnecessary details that are irrelevant to the review topic. For example, Section 2 - Lines 77-81 and 93-98- What is the relevance of these details with this review topic? Consider summarizing or omitting these details to maintain focus.

7.      Section 3 - Plant Salt Stress and Saline Land Management – similar to the above comment- What is the relevance of sections 3.2 and 3.3. with this review topic?

8.      Scientific names of organisms (plants, bacteria) should be in italics (Line 337, 481). Authors should carefully check the whole text for this issue.

Comments on the Quality of English Language

Extensive editing of English language required

Author Response

Revision list according to the comments from Reviewer 1

Guo et al. have reviewed the potential use of plant endophytes to deal with the salinity stress in plants grown in salt-alkali soils. However, the manuscript requires substantial improvements for clarity and readability.

Response: Many thanks for your comments and suggestions. We have re-written the whole manuscript and figure 1&2 have been re-graphed. The MS language have been edited by Editage.

  1. What do the authors mean by salt-alkali soil? No information is provided, and this term is not consistently used in the manuscript. This term needs clarification and consistent usage throughout the manuscript.

Response: Thank you for your comment. We have added the description of salt-alkali soil in Lines 27-31 in the revised version and revised this term in the manuscript.

  1. The manuscript contains sentences that lack clarity and may appear to be AI-generated. Extensive language editing is required to enhance readability. For example: For example,

Line 26: "However, large tracts of land are wasted…" -> Rewrite for clarity.

Line 61-62: "In addition to traditional treatments such as physical, chemical, and agronomic improvements by plants…" -> Clarify the meaning.

Line 309: "Healthy to wilted plants after receiving salt-alkali stress?" -> Rewrite for clarity.

Line 546: "Endophytes are an emerging method for salinity treatment?" -> Clarify the statement.

Response: Thanks for your concern, and revised. The MS language have been edited by Editage.

  1. Some of the text needs reference. Authors should carefully check the whole text for this issue.

I am providing a couple of examples-

Line 43- Sunflowers, licorice, alfalfa, and other salt-tolerant plants can be planted (provide reference). Endophytes may contribute to the chlorophyll content of the plant under salt stress, effectively increasing their photosynthetic capacity (provide reference).

Response: Thanks for all your concerns. Based on your comments, we have deleted sections 3.2 to 3.4, including 13 references, and then added 18 references into the relevant sentences.

  1. All the figures are too general and do not provide any insights into the topic. Authors should refer to the relevant literature and update the figures with more meaningful content. For instance, Figure 1 and Figure 2 should include an overview of salinity and alkali stress sensing, signaling, and response in plants. For instance, Figure 2 is primitive, and several aspects of salinity stress are missing.

Response: Thanks for all your suggestions. The Figure 1 and 2 have been re-graphed and the Figure 3 and 4 have been re-arranged.

  1. Also, Figure legends are inadequate and ensure that figure legends provide sufficient information for clarity.

Response: Thanks, revised as suggested.

  1. Some sections contain unnecessary details that are irrelevant to the review topic. For example, Section 2 - Lines 77-81 and 93-98- What is the relevance of these details with this review topic? Consider summarizing or omitting these details to maintain focus.

Response: Done as suggested, thanks.

  1. Section 3 - Plant Salt Stress and Saline Land Management – similar to the above comment- What is the relevance of sections 3.2 and 3.3. with this review topic?

Response: Thanks for all your concerns. We have deleted the sections 3.2, 3.3 and 3.4, rewritten the Section 3.

  1. Scientific names of organisms (plants, bacteria) should be in italics (Line 337, 481). Authors should carefully check the whole text for this issue.

Response: Thanks, revised as suggested.

Reviewer 2 Report

Comments and Suggestions for Authors

The manuscript entitled “The Potential of Endophytes in Relieving Salt-alkali Tolerance and Improving Salinity Resistance in Plants” discusses the potential beneficial effects of Endophytes in the management of plant growth in saline-alkali soil and improve salinity resistance.

In recent times, plant-associated endophytes are emerging candidates in multifaceted socio-economic applications, including the potential of these beneficial microbes in alleviating biotic/abiotic stress and phytoremediation and several endophyte strains are extensively studied for these attributes.

The manuscript offers interesting insights into how endophytes can be utilized to alleviate salinity stress in plants and confer resistance against high salt concentrations and represents a potential area of research.

Studies have highlighted multiple roles of endophytes and need not be discussed much in the manuscript. The authors should provide clear information on how endophytes alleviate salinity stress and their mechanisms.

Although the topics of the manuscript are of interests, the article is poorly written and framed and has to be extensively revised with a focus on the title of the manuscript, random information is not required.

Subtopic 2. Plant Endophytes can be written as “Plant-associated Endophytes”

Line 106-107. Currently, endophytes are used as receptors to transfer disease or insect resistance genes…the sentence should be rewritten.

Line 110-111. Overall, endophytes have drastically changed our way of life and production, providing plenty of ideas on how to alleviate salt stress in plants. What does it mean?

Line270-271. Endophytes originate from saline environments and can eventually be released back into nature to improve the saline environment???

The sentence should be reframed as “Endophytes inhabiting plants in saline environments can be isolated and used in bioremediation of saline conditions”.

English language needs to be extensively improved as well as the sub-topics of the manuscript.

Line 337. The scientific name Salicornia salsa should be in italics.

In many sections, random information is presented, e.g. salinity tolerance of plants and bacteria, it is not needed.

Table 1, Spelling of Endophyte needs to be corrected, likewise throughout the text.

References can be improved.

Comments on the Quality of English Language

Extensive improvement of English Language is required.

Author Response

Revision list according to the comments from Reviewer 2

Although the topics of the manuscript are of interests, the article is poorly written and framed and has to be extensively revised with a focus on the title of the manuscript, random information is not required.

Response: Thanks for your comment and suggestions. We have revised the MS and the language have been edited by Editage.

Subtopic 2. Plant Endophytes can be written as “Plant-associated Endophytes”

Response: Done as suggested, thanks.

Line 106-107. Currently, endophytes are used as receptors to transfer disease or insect resistance genes…the sentence should be rewritten.

Response: Thanks, the sentence has been rewritten.

Line 110-111. Overall, endophytes have drastically changed our way of life and production, providing plenty of ideas on how to alleviate salt stress in plants. What does it mean?

Response: Thanks for your comments. Based on your comments, we have rewritten the sections 2.

Line270-271. Endophytes originate from saline environments and can eventually be released back into nature to improve the saline environment???

Response: This sentence has been revised “Endophytes inhabiting plants in saline environments can be isolated and used in the bioremediation of land damaged by high-salinity conditions.”  

The sentence should be reframed as “Endophytes inhabiting plants in saline environments can be isolated and used in bioremediation of saline conditions”.

Response: Done as suggested, thanks.

English language needs to be extensively improved as well as the sub-topics of the manuscript.

Response: Thanks for your concern, and revised. The MS language have been edited by Editage.

Line 337. The scientific name Salicornia salsa should be in italics.

Response: Thanks for your concern, and revised.

In many sections, random information is presented, e.g. salinity tolerance of plants and bacteria, it is not needed.

Response: Thanks for your concern, and revised.

Table 1, Spelling of Endophyte needs to be corrected, likewise throughout the text.

Response: Thanks for your concern, and corrected.

References can be improved.

Response: Thanks for your concern, and the References have been improved.

Round 2

Reviewer 2 Report

Comments and Suggestions for Authors

The authors have revised the manuscript as per suggestions. 

The manuscript may be considered.